# High-Risk Pedigree Study Identifies *LRBA* (rs62346982) as a Likely Predisposition Variant for Prostate Cancer

**DOI:** 10.3390/cancers15072085

**Published:** 2023-03-31

**Authors:** Lisa A. Cannon-Albright, Jeff Stevens, Julio C. Facelli, Craig C. Teerlink, Kristina Allen-Brady, Neeraj Agarwal

**Affiliations:** 1Genetic Epidemiology, Department of Internal Medicine, University of Utah School of Medicine, Salt Lake City, UT 84132, USA; 2George E. Wahlen Department of Veterans Affairs Medical Center, Salt Lake City, UT 84148, USA; 3Huntsman Cancer Institute, University of Utah, Salt Lake City, UT 84112, USA; 4Department of Biomedical Informatics and Clinical and Translational Science Institute, Spencer Fox Eccles School of Medicine, University of Utah, Salt Lake City, UT 84112, USA; 5Division of Oncology, Department of Internal Medicine, University of Utah School of Medicine, Salt Lake City, UT 84132, USA

**Keywords:** prostate cancer, predisposition, high-risk pedigree, UPDB, *LRBA*

## Abstract

**Simple Summary:**

There remains a lack of identification of genes responsible for the most aggressive of prostate cancers. A unique resource consisting of sampled, related men who died from their confirmed prostate cancer and who were members of pedigrees shown to have a significant excess of prostate cancer cases has allowed the identification of multiple rare candidate predisposition variants for prostate cancer. Strong candidate variants are presented, including a rare, validated variant in the gene *LRBA*, all of which can be examined further in additional studies.

**Abstract:**

There is evidence for contribution of inherited factors to prostate cancer, and more specifically to lethal prostate cancer, but few responsible genes/variants have been identified. We examined genetic sequence data for 51 affected cousin pairs who each died from prostate cancer and who were members of high-risk prostate cancer pedigrees in order to identify rare variants shared by the cousins as candidate predisposition variants. Candidate variants were tested for association with prostate cancer risk in UK Biobank data. Candidate variants were also assayed in 1195 additional sampled Utah prostate cancer cases. We used 3D protein structure prediction methods to analyze structural changes and provide insights into mechanisms of pathogenicity. Almost 4000 rare (<0.005) variants were identified as shared in the 51 affected cousin pairs. One candidate variant was also significantly associated with prostate cancer risk among the 840 variants with data in UK Biobank, in the gene *LRBA* (*p* = 3.2 × 10^−5^; OR = 2.09). The rare risk variant in *LRBA* was observed to segregate in five pedigrees. The overall predicted structures of the mutant protein do not show any significant overall changes upon mutation, but the mutated structure loses a helical structure for the two residues after the mutation. This unique analysis of closely related individuals with lethal prostate cancer, who were members of high-risk prostate cancer pedigrees, has identified a strong set of candidate predisposition variants which should be pursued in independent studies. Validation data for a subset of the candidates identified are presented, with strong evidence for a rare variant in *LRBA*.

## 1. Introduction

There is evidence for an inherited contribution to predisposition to prostate cancer, and stronger evidence for lethal prostate cancer [1,2,3,4,5,6]. While hundreds of common variants in low-risk genes have been associated with risk for prostate cancer (e.g., [7]), few rare variants in moderate- to high-penetrance genes have been identified, and they explain little of familial prostate cancer [8,9]. The genes most consistently recognized to affect prostate cancer risk include *ATM*, *BRCA1*, *BRCA2*, *CHEK2*, *HOXB13*, *MLH1*, *MSH2*, *MSH6*, *PMS2*, and *PALB2* [10], not all of which display equivalent penetrance.

We sequenced germline DNA from 51 pairs of cousins who both died from prostate cancer who were also members of pedigrees with a significant excess of prostate cancer cases. We identified the set of rare coding and noncoding variants shared in these high-risk lethal prostate cancer-affected cousin pairs as strong candidate variants for predisposition to prostate cancer. Confirmation of significant association with prostate cancer risk in an independent population and observation of segregation of variants with prostate cancer in multiple high-risk pedigrees provided additional validation for a subset of the candidates considered. We used 3D protein structure prediction methods to analyze structural changes in one outstanding candidate variant that may provide insights on mechanisms of pathogenicity [11].

## 2. Data/Methods

### 2.1. Utah Population Database (UPDB)

The UPDB consists of the genealogy of the majority of the Utah population, from its founders in the mid-1800s to their modern-day descendants. The genealogy has been linked to the Utah Cancer Registry (UCR) and to Utah death certificates coded with International Classification of Disease causes of death from 1904. The statewide UCR was created in 1966 and has been an NCI Surveillance, Epidemiology, and End-Results (SEER) registry since 1973. It records and tracks all independent primary cancers diagnosed or treated in Utah, including pathologic confirmation. Of the over 3 million individuals with genealogy of at least 3 generations linking to Utah founders, there were 7727 individuals whose linked Utah death certificate indicated prostate cancer as a cause of death; 6328 of these individuals also had a UCR record confirming a prostate cancer diagnosis.

Using the combined UPDB genealogy and UCR data, the Genetic Epidemiology Program at the University of Utah has previously identified and sampled ~2500 prostate cancer cases and ~7500 relatives who belonged to approximately 500 Utah pedigrees that each had an excess of prostate cancer cases among the descendants (high-risk pedigrees). In total, 422 of these sampled prostate cancer cases were since identified to have died from their prostate cancer based on the presence of prostate cancer as a cause of death on their linked Utah death certificate. These men were termed lethal prostate cancer (LPrCa) cases. All genetic relationships among these cases were identified in the UPDB genealogy to identify descending pedigrees.

### 2.2. Affected LPrCa Cousin Pairs in High-Risk Pedigrees 

All independent descending clusters (pedigrees) including 2 or more of the 422 sampled men who died of their prostate cancer (LPrCa cases) were identified in the UPDB. No pedigrees were completely overlapping, but an LPrCa case could belong to more than one pedigree through different ancestors. Each pedigree that included at least two sampled LPrCa cases was tested for a significant excess of prostate cancer cases as follows. All individuals with biological sex of male and at least three generations of genealogy linking to Utah founders were assigned to a cohort based on five-year birthyear and birth state (Utah or not). Prostate cancer cohort-specific rates were estimated as the number of prostate cancer cases in each cohort divided by the number of males with genealogy data in the cohort. Each pedigree was tested for an excess of prostate cancer cases by comparing the observed number of prostate cancer cases among the descendants to the expected number of prostate cancer cases among the descendants. The expected number of cases among the descendants was estimated by summing the cohort-specific rate of prostate cancer for all male descendants. A pedigree was termed high-risk for prostate cancer in the presence of a significant excess of observed cases (*p* < 0.05). For each of these high-risk prostate cancer pedigrees identified, which also contained at least 2 sampled LPrCa cases, we selected those LPrCa cases who were related as first or second cousins for sequencing. Each of the 102 LPrCa cases in the 51 cousin pairs identified had a stored sample of germline DNA extracted from whole blood available for whole-genome sequencing.

### 2.3. Whole-Genome Sequencing/Identification of Candidate Predisposition Variants/Assay Development

The 102 samples of high-molecular-weight DNA (>30 Kb in length) were whole-genome sequenced at MedGenome, Foster City, CA, USA. (http://www.medgenome.com) utilizing 10× Genomics (https://www.10xgenomics.com/) long, linked read sequence technology. A high efficiency microfluidic device mixes functionalized gel beads containing unique barcodes with enzymes and a limiting amount of genomic DNA. These components are encapsulated in oil to produce a GEM, gel bead in emulsion. With up to 4 million barcodes available, GEM particles produce uniquely identifiable high-molecular-weight DNA fragments. These fragments are then sequenced utilizing Illumina sequencers to produce 30× whole-genome phased coverage. Genomes were aligned to GRCh37 with 10× Genomics LongRanger software version 2.2.2. VCFs were merged with BCFTOOLS. The merged VCF was annotated with ANNOVAR ([12]; https://academic.oup.com/nar/article/38/16/e164/1749458).

A total of 114,513 rare (MAF < 0.005 in GnomAD 2.1) coding (nonsynonymous, frameshift, startloss, startgain, stoploss) variants were identified in the 102 LPrCa cases. Of these, 17,859 variants had allele counts >1, and 3251 of these rare coding variants (in 1762 genes) were shared within at least 1 LPrCa cousin pair. Noncoding and UTR variants were prioritized after RegulomeDB scoring. The RegulomeDB score represents evidence that each variant functions in a regulatory role (ranging from 1, indicating strong evidence, to 6, indicating weak evidence). A total of 546 rare noncoding variants with RegulomeDB scores ranging between 2a and 4, and shared in at least 1 cousin pair, were identified. 

These 3797 (3251 coding + 546 noncoding) rare, shared candidate predisposition variants were submitted for assay design using the Illumina iSelectHD Custom Genotyping BeadChips (https://www.illumina.com/products/by-type/informatics-products/designstudio.html). A total of 200 base pairs of genomic sequence surrounding each variant was used for the design process. Assays were manufactured for 3559 of these candidate variants (238 failed to design). Illumina iSelectHD Custom Beadchips were processed at the University of Utah HSC Genomics Core.

### 2.4. Segregation of Candidate Predisposition Variants in Pedigrees

In total, 1298 additional previously sampled Utah prostate cancer cases were identified for assay with the candidate predisposition variants to determine segregation of the candidate variants. These assayed prostate cancer cases included the original affected cousin pairs (n = 102); all other sampled prostate cancer cases whose death certificate indicated prostate cancer as a cause of death (n = 320 LPRCA cases); all sampled prostate cancer cases who were first-, second-, or third-degree relatives of the affected cousin pair cases (n = 168) and all sampled prostate cancer cases who were first-, second-, and third-degree relatives (n = 307) of these 168 cases; and all metastatic prostate cancer cases recruited by author N.A. in the Huntsman Cancer Institute Urology Clinic (n = 401). DNA for 1195 of these 1298 sampled prostate cancer cases passed quality control and was assayed for the candidate variants to test for segregation and to identify additional carriers.

### 2.5. Case/Control Risk Association in UK Biobank

These rare, shared candidate predisposition variants that were identified in the sequencing experiment and had data in UK Biobank were analyzed for association with prostate cancer risk in a set of 7764 Caucasian prostate cancer cases and 1:1 ancestrally matched controls from among the UK Biobank’s 488,377 total subjects genotyped on the Illumina OmniExpress SNP array [13]. UK Biobank case and control subjects were matched via principal components (PC) using ~27K independent markers that excluded several genomic regions known to adversely affect PC analysis [14]. FLASHPCA2 software was used to generate eigenvectors for control selection [14]. Controls were selected from among 64,284 Caucasian UK Biobank subjects who were male, over 70 years of age, and had no cancer diagnosis. One control, representing the nearest neighbor based on the Euclidean distance of the first two PCs, was selected for each case. Then, 129 outlier cases and controls were removed, leaving 7635 cases and controls. 

### 2.6. UK Biobank Imputation 

The selected UK Biobank case and control subjects were imputed to ~40M SNP markers using the Haplotype Reference Consortium’s (HRC) 67 K background genomes [15]. Beginning with 784,256 observed SNP genotypes, pre-imputation quality control using PLINK software [16] required sample genotyping > 98% (no subjects removed); a total of 353,578 markers were removed by filtering for genotyping call rate < 98%, HWE *p* < 1 × 10^−5^, MAF < 0.005, duplicated position in the HRC’s reference genome, or site not included in the HRC’s reference genome. The remaining 430,678 SNPs were converted to human genome B37 forward strand orientation using GenotypeHarmonizer software [17] and served as the basis for imputation. Imputation was performed with EAGLE v2.3 software for phasing [18] and MINIMAC3 software for imputation [19]. Post-imputation quality control included removing markers with imputation information score (INFO-r^2^) < 0.7 [7,20,21]. 

### 2.7. Protein Structure Prediction

LRBA (lipopolysaccharide-responsive beige-like anchor protein) is a very large protein with 2863 amino acids that is not tractable using existing 3D structure prediction methods. The only experimental structure available for LRBA is an X-ray structure for positions 2076–2489 [22,23]. There are two isoforms; here, we consider the isoform 1 as the canonical sequence. For this sequence, the single point mutation considered here is T2533P. The protein has multiple domains, and the mutation in question, T2533P, is localized between the second BEACH (2200–2489) domain and the second WD2 (2591–2633) domain. To perform 3D structure prediction with I-TASSER structure prediction software using full homology modeling [24,25,26,27], we constructed a model including the PH domain (2073–2181), the BEACH domain (2200–2489), and the WD2–WD6 domains (2591–2858). For this model, the amino acid substitution under consideration is T461P. The I-TASSER predictions resulted in a set of three candidate structures with C-scores of −3.86, −3.85, and −3.93 for the wild-type sequence, and two with C-scores of −3.74 and −3.80 for the mutant sequence. All the structures obtained were visualized and analyzed using Chimera [28]. 

## 3. Results

### 3.1. Case/Control Risk Association

Only 840 of the 3797 candidate predisposition variants had some data present in UK Biobank, likely due to the very low frequency (<0.005). Only 1 of the 840 variants was significantly associated with prostate cancer risk after correction for multiple testing (*p* < 0.05/840 or *p* < 5.95 × 10^−5^), in the gene *LRBA* (*p* = 3.2 × 10^−5^; OR = 2.09).

### 3.2. Assay of Candidate Variants in 1195 Prostate Cancer Cases

In total, 102 of the 1195 prostate cancer cases assayed for the candidate variants were the original target pair LPrCa cases; these cases were assayed to validate performance of the assays. Of the 3559 variants assayed, 2035 failed to identify any heterozygous (het) carriers with the assay. These variants were considered to have failed performance, since the assay should have at least identified the original cousin pair carriers. For 562 of the variants, although some het carriers were identified, the assay did not identify the original cousin pair as carriers, so these variants were also considered to have failed the assay. For 78 of the variants, the only carriers identified in the assay were the original cousin pair; these variant assays were considered to have performed appropriately, but they did not identify any additional carriers of the candidate variants. This finding was not unexpected given the rare nature of the variants, but these variants did not provide any additional evidence for segregation and were not further considered here. For the remaining 884 of the variant assays, all original cousin pair carriers were identified, as well as additional het carriers. These 884 candidate variants were considered the best rare candidate lethal prostate cancer predisposition variants. These variants were examined further in the 1195 prostate cancer cases assayed, to determine if segregation to other related prostate cancer cases was observed.

All het carriers for each of these 884 rare candidate variants were analyzed by identifying all genealogical relationships among the het carriers using the UPDB linked genealogy. This analysis identified many more clusters than the original 51 pedigrees analyzed since all genetic relationships among all cases were considered; 1959 total clusters (pedigrees) that included 3 or more related het carriers for at least 1 of these 884 variants were identified. Although none of these pedigrees overlapped completely, a case could belong to more than one. 

These 884 candidate variants were all originally selected for their low frequency (<0.005) based on review of GnomAD. However, some of the variants were observed at much higher frequency than 0.005 in the assay of 1195 prostate cancer cases. It is possible that these variants might represent truly rare variants that were observed in much higher frequencies in the assay of prostate cancer cases because they are, in fact, strongly associated with prostate cancer risk. However, these results might also be due to inaccuracies in reporting of frequencies for what were, in fact, more common variants, or due to assay performance failure. We could not confirm the lack of a data problem for these more commonly observed variants, so we elected to exclude consideration of any of the 1959 variant-segregating pedigrees for any variant for which more than 10% (n > 110 carriers) het carriers were observed in the assay of 1195 prostate cancer cases. 

This left 1070 pedigrees of interest that included at least 3 related het carriers of 1 of the candidate variants. Because some of these clusters of het carriers might represent random clusters rather than high-risk prostate cancer pedigrees, we additionally excluded those variant-sharing pedigrees that did not exhibit a significant excess of prostate cancer cases (*p* < 0.05) among the descendants of the common founding ancestor of all the related carriers. This left 814 high-risk prostate cancer pedigrees, including 559 different rare candidate variants in 398 genes that showed evidence of segregation of the candidate variant with prostate cancer. The list of these 559 rare candidate predisposition variants is provided in Appendix A, which includes the UK Biobank association test results for those variants with data, the diagnosis and interpretation data from ClinVar, and identifies those variants associated with a prostate cancer pathway in Ingenuity. 

Only 1 of these 559 candidate variants was significantly validated for risk association with prostate cancer (*LRBA*). This *LRBA* variant (rs62346982) is classified in ClinVar with “conflicting interpretations of pathogenicity”. The *LRBA* variant was observed to segregate with prostate cancer in a total of five high-risk prostate cancer pedigrees. Figure 1 shows segregation of the rare *LRBA* variant in the largest *LRBA*-segregating pedigree identified. The founder of this pedigree was born in the late 1700s in Vermont and has almost 32,000 descendants in UPDB, with a total of 230 prostate cancer cases observed and 174.9 expected (*p* = 4.0 × 10^−5^) among all descendants (additional UPDB data not shown); variant-carrying cases are shown. A multiple marriage in the third generation is noted with a triangle on the two horizontal marriage lines, variant carriers are noted by “+”, and the decade of prostate cancer diagnosis is noted below each case. It must be noted that complete prostate cancer case diagnosis data for Utah are only available from 1973, which explains why cases are only observed in the bottom, most recent few generations of the high-risk pedigrees shown.

### 3.3. Protein Structure Prediction

Lipopolysaccharide-responsive and beige-like anchor protein (*LRBA*), the single candidate predisposition variant identified in a cousin pair and also confirmed with significant association to prostate cancer in the UK Biobank case/control analysis, is involved in coupling signal transduction and vesicle trafficking to enable polarized secretion and/or membrane deposition of immune effector molecules (this function was assigned by similarity). It is involved in phagophore growth during mitophagy by regulating ATG9A trafficking to mitochondria. A mutation of this protein has been associated with CVID8, an autosomal recessive immunologic disorder associated with defective B-cell differentiation.

The variant considered here corresponds to rs62346982. PolyPhen-2 predictor predicts this variant as benign with a score of 0.167 (sensitivity 0.92 and specificity 0.87) [29]. The overall predicted sequences do not show any significant changes upon the amino acid replacement T461P in the model considered here, but careful inspection of the region close to the replacement (440–480) shows that the mutated structure loses a helical structure two residues after the replacement (see Figure 2 and Figure 3). The water accessibility does not change upon mutation, keeping this region buried.

### 3.4. Consideration of Other Likely Candidate Predisposition Variants Identified

In addition to the significant evidence identifying the rare *LRBA* variant as a strong candidate prostate cancer predisposition variant, this study identified a large set of other rare candidate predisposition variants, many of which are worthy of further consideration. We have used various criteria to select a subset of these LPrCa candidate predisposition variants for presentation in more detail below. These criteria include: (i) known cancer pathogenic variants, (ii) variants in genes recognized to be implicated in breast cancer (BROCA genes), and (iii) noncoding variants with suggestive Regulome DB scores.

(i)Cancer pathogenic variants

Only one candidate variant that was classified as “pathogenic/likely pathogenic” in ClinVar was identified, in *MUTYH* (seq_1_45797228_C-T_MUTYH). This variant was identified in 29/1195 of the assayed cases and segregated in 3 high-risk prostate cancer pedigrees including 6, 3, and 3 sampled related cases, respectively. Figure 4 below shows the largest of these three pedigrees, with full shading for prostate cancer cases; the six assayed het carriers are identified with “+”. One inferred case carrier (not sampled) is also shown; decade of age at diagnosis of prostate cancer is shown below each case. The founder of this pedigree was born in New York in the early 1800s, with a total of almost 16,000 descendants included in UPDB; only the descending lines to the 6 identified het carriers are shown. In the UPDB, a total of 114 prostate cancer cases were observed among all descendants, with 68.3 cases expected (*p* = 2.7 × 10^−7^). The founding male had two marriages, indicated with a small triangle on the top horizontal marriage lines.

Eleven candidate variants found to segregate in high-risk pedigrees were classified with “conflicting interpretations of pathogenicity” in ClinVar; these included variants in *DUOX2*, *LRBA* (discussed previously), *MYO3A*, *EVC*, *CP*, *FBN1*, *MYH11*, *KCND3*, *ACNT2*, *ADAM9,* and *PTCH1.* The largest pedigree observed for the “conflicting” variant identified for *DUOX2* (n = 14 case carriers) is shown in Figure 5, and the largest pedigree observed for *LRBA* (n = 10) was shown in Figure 1. *PTCH1* was also selected as a BROCA gene, and a pedigree is shown later. A variant in *MSH6* was the only variant of 14 classified in ClinVar as of “uncertain significance” that also reported a cancer-related clinical diagnosis; it was also selected as a BROCA variant and is shown later.

Figure 5 shows the segregation of the rare *DUOX2* variant in a high-risk prostate cancer pedigree. The founder of this pedigree was born in the mid-1700s in North Carolina and has almost 153,000 descendants in UPDB; 897 prostate cancer cases were observed among all descendants, with 811.5 expected (*p* = 0.0016). The male founder of this pedigree had two spouses, shown with a triangle on the top horizontal marriage lines. All 14 prostate cancer case variant carriers are shown with “+”, and 1 inferred case (father of an affected carrier who was not sampled) is also shown.

(ii)Variants in recognized BROCA genes

Seven of the rare candidate predisposition variants found to segregate in prostate cancer cases who were members of high-risk prostate cancer pedigrees were in known BROCA genes, including the *MUTYH* variant already discussed, as well as variants in *APC* (ClinVar: conflicting interpretations of pathogenicity), *BRCA1* (benign), *MSH6* (uncertain significance), *PTCH1* (conflicting), *SDHC* (not classified), and *SLX4* (benign/likely benign). The pedigrees segregating the rare variants in *BRCA1*, *MSH6,* and *PTCH1* are shown in Figure 6, Figure 7 and Figure 8. 

Figure 6 shows the single high-risk prostate cancer pedigree segregating a *BRCA1* variant (seq_17_41228587_T-G_BRCA1). This variant has been classified as “benign” in ClinVar, based on the breast cancer phenotype. The founder of this pedigree was born in the early 1800s in Canada and has a total of over 2300 descendants in the UPDB; a total of 25 prostate cancer cases were observed among all descendants, with 12.1 expected (*p* = 7.6 × 10^−4^). The three identified variant carriers are shown with “+”; also shown is the unsampled pedigree member father of one of the affected pair cases who was also diagnosed with prostate cancer. This pedigree also has a significant excess of breast cancer (18 observed, 11.3 expected, *p* = 0.04). There are no identified breast cancer-affected descendants of the first son of the founder pair shown in Figure 6, but the second son of the founder has eight descendants diagnosed with breast cancer (UPDB breast cancer cases not shown).

Figure 7 shows the single high-risk pedigree segregating a rare *MSH6* variant (seq_2_48026979_A-C_MSH6). This variant is classified as “uncertain” in ClinVar. The founder of this pedigree was born in the early 1800s in Massachusetts and has almost 23,000 descendants in UPDB; 166 prostate cancer cases were observed among all descendants, with 113.0 expected (*p* = 1.8 × 10^−6^); 129 breast cancers were observed with 109.8 expected (*p* = 0.04); 23 ovarian cancers were observed with 13.9 expected (*p* = 0.016). Neither colorectal nor endometrial cancers were observed in excess among the descendants in this pedigree. Prostate cancer case variant carriers are shown with “+”; one unsampled father of a carrier was also diagnosed with prostate cancer as shown. Two of the prostate cancer case carriers had an additional cancer diagnosis in UPDB (one kidney and one chronic lymphocytic leukemia); data are not shown to protect privacy.

Figure 8 shows the single high-risk pedigree segregating a rare *PTCH1* variant (seq_9_98240378_C-T_PTCH1). This variant is classified as “conflicting” in ClinVar. The founder of this pedigree was born in the mid-1800s with no recorded birthplace and has almost 4600 descendants in UPDB; a total of 33 prostate cancers were observed among all descendants, with 20.0 expected (*p* = 0.0047). This pedigree also exhibits a significant excess of breast cancer: a total of 37 breast cancers were observed in the entire descending pedigree, with 23.1 expected (*p* = 0.0045). One of these breast cancer cases was the mother of a carrier, shown with half shading.

(iii)Noncoding variants with suggestive Regulome DB scores

Regulome DB (RGDB) scores ranging from “2a” to “4” were observed among the rare candidate variants. The two variants with the most evidence for a regulatory role based on RGDB score were in *PAX6* and *ZWILCH*, and both were scored “2a”. Two pedigrees with the *PAX6* variant were observed with three het carriers each, and one pedigree with the *ZWILCH* variant also included three het carriers; these pedigrees are not shown. 

Multiple variants with an RGDB score of “2b” were identified, but pedigrees are not shown here. The founder of the largest pedigree, with 11 het carriers of an *APCDD1L-DT* variant with score “2b”, was born in the late 1700s in Massachusetts and has over 36,000 total descendants, with 268 prostate cancer cases observed among all descendants, and 167.8 expected (*p* = 6.6 × 10^−13^). 

## 4. Discussion

It is estimated that 10–20% of prostate cancer cases occur in a familial context [1,30]. Genetic predisposition to prostate cancer development has been associated with both rare variants in moderate- to high-penetrance genes, as well as with common genetic alterations in low-risk genes (e.g., [7,8]). Almost 200 common variants have been identified in large case–control cohorts, but other evidence for their direct association with prostate cancer is lacking, and only a small fraction of familial prostate cancer is associated with known rare predisposition variants. 

High-risk pedigree studies remain a powerful mechanism for identification of predisposition genes and variants [31,32,33,34]. This has proven true for prostate cancer [35,36], although such high-risk prostate cancer pedigrees remain infrequently presented. Here, we have taken advantage of unique Utah resources, combined with an unusual and powerful study design that includes sampled affected cousin pairs, to generate, and begin to evaluate, a strong list of candidate predisposition variants for prostate cancer.

Extensive linked genealogic and disease registries existing in Utah have been used to identify and study thousands of Utah high-risk pedigrees [37]. We have previously used this same sequencing approach in affected cousins belonging to high-risk pedigrees to identify multiple candidate predisposition variants for several different phenotypes, including *GOLM1* for melanoma [38], *ERF* for bladder cancer [36], *FANCM* for colorectal cancer [39], *MEGF* for osteoporosis [40], *HOXC4* for Chiari Malformations [41], and multiple candidates for Alzheimer’s [42] and exceptional longevity [43], among others. 

Using the combined genealogy and cancer registry resource in the UPDB, the strongest evidence for an inherited contribution to prostate cancer has been shown for the subset of lethal prostate cancer cases [44]. The present analysis of available germline DNA for 51 pairs of cousins who each died of prostate cancer and belonged to a pedigree with a significant excess of prostate cancer among the descendants has identified thousands of rare, shared coding and noncoding variants in those cousins, each of which represents a candidate predisposition variant for prostate cancer which can be explored further. Since validation of segregation and risk association was based on the prostate cancer phenotype, rather than on the rarer (and remaining unknown for most cases) phenotype of lethal prostate cancer, conclusions may therefore be limited to prostate cancer rather than lethal prostate cancer, pending further study.

One outstanding candidate, in *LRBA*, was the only candidate predisposition variant to be further validated in the independent UK Biobank population for association with prostate cancer risk; this variant was found to segregate in five independent Utah high-risk pedigrees. While protein prediction modeling for this variant did not show any significant changes, the mutated structure does lose a helical structure two residues after the replacement (Figure 2 and Figure 3). The COACH results did not find any binding site in the region considered for this protein, nor is there any ligand information from experimental studies; therefore, it is not clear how to relate the additional helix structure with possible increase of loss of function of the protein [45,46]. Further study of the specific Utah high-risk pedigrees identified here, as well as analysis in other independent populations of prostate cancer cases, can further clarify whether, and how, this variant is specifically associated with increased risk for prostate cancer or lethal prostate cancer.

While not all the thousands of rare, shared candidate prostate cancer predisposition variants identified in this study have been reviewed in detail due to limitations of time and space, several specific subsets of variants were presented in more detail. Subsets considered included known cancer pathogenic variants, BROCA genes, and noncoding variants with suggestive Regulome DB scores. The identification of multiple strong candidate variants in these subsets, some of which represented more than one subset (e.g., *PTCH1* or *MSH6*) demonstrates the power of this approach and serves as validation of this approach to identify both known and new predisposition variants.

The strengths of this study include the large number of affected cousin pairs analyzed, leading to many rare, shared candidate predisposition variants being identified. Strength also comes from the analysis of the phenotype of lethal prostate cancer in the proband pairs, which was confirmed both by a linked Utah death certificate for prostate cancer and validation of the cancer diagnosis within the linked Utah Cancer Registry. The available stored germline DNA from thousands of Utah prostate cancer cases, many of whom were related to the affected cousin pairs sequenced, allowed further validation of candidate variants by the confirmation of segregation of the variants. Limitations of this study include the censoring of some death, cancer, and genealogy data, which may have occurred through errors of reporting or record linking, or through lack of records. Also of note is that the founding population of Utah is largely northern European [47], and while immigration has been significant, these results may only specifically apply to this limited population and will require confirmation in other independent and more diverse populations. Confirmation of these results for the lethal prostate cancer phenotype will have to be made when more data for this more extreme phenotype are available in other independent populations.

## 5. Conclusions

Analysis of high-risk cancer pedigrees identified in this powerful Utah resource previously provided the identification of *BRCA1* and *BRCA2* [48,49], and *CDKN2A* [50], which remain the most common cancer predisposition genes to be identified. Large-scale case and pedigree studies strongly suggest that other common cancer predisposition genes may not remain to be identified and that, rather, most familial cancer predispositions might be the result of many varied, rare predisposition genes and variants [34]. Whether or not this is the case, studies such as this one, which analyze related affected individuals within a large number of high-risk pedigrees, have shown the strong potential to identify many candidate predisposition genes and variants for many different phenotypes. Such studies should be pursued, and the candidate predisposition variants identified are worthy of further exploration.

## Figures and Tables

**Figure 1 cancers-15-02085-f001:**
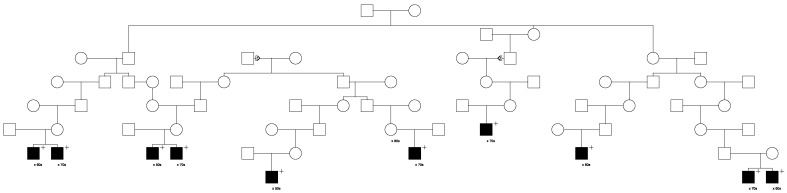
One of the high-risk prostate cancer pedigrees segregating the rare *LRBA* candidate prostate cancer predisposition variant.

**Figure 2 cancers-15-02085-f002:**
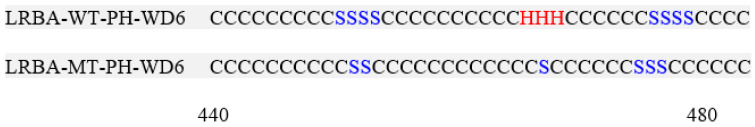
Secondary structure comparison in the region of the T461P replacement.

**Figure 3 cancers-15-02085-f003:**
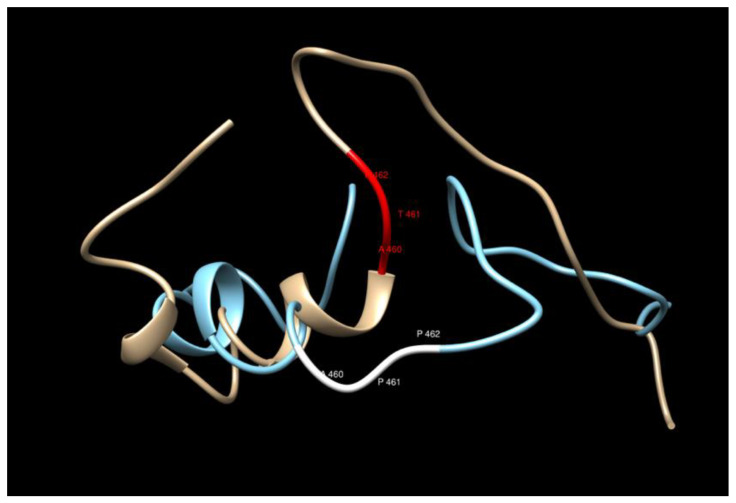
The 3D structure comparison in the region of the T461P replacement. The bronze/red structure corresponds to the WT and the blue/white one to the MT.

**Figure 4 cancers-15-02085-f004:**
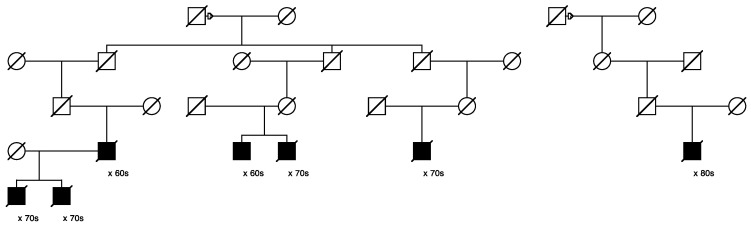
High-risk prostate cancer pedigree segregating the rare pathogenic *MUTYH* variant.

**Figure 5 cancers-15-02085-f005:**
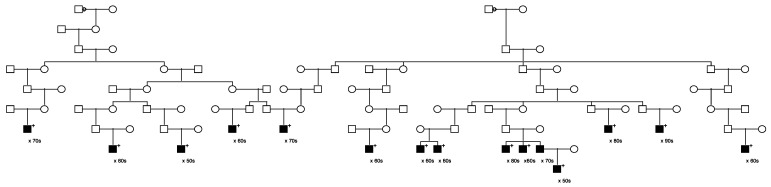
High-risk prostate cancer pedigree segregating a rare *DUOX2* variant.

**Figure 6 cancers-15-02085-f006:**
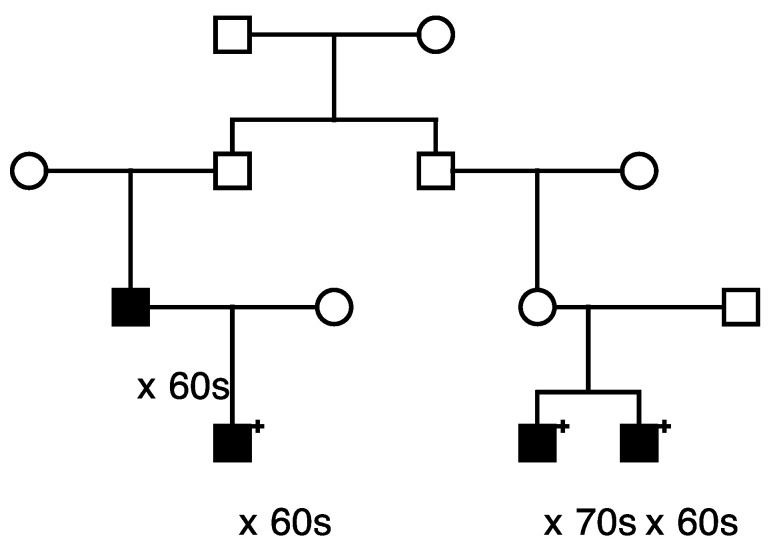
High-risk prostate cancer pedigree segregating a rare BRCA1 variant.

**Figure 7 cancers-15-02085-f007:**
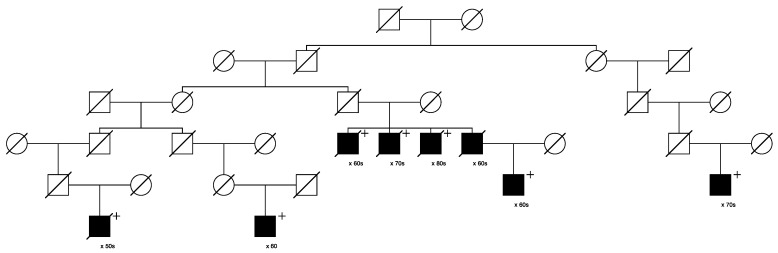
High-risk prostate cancer pedigree segregating a rare *MSH6* variant.

**Figure 8 cancers-15-02085-f008:**
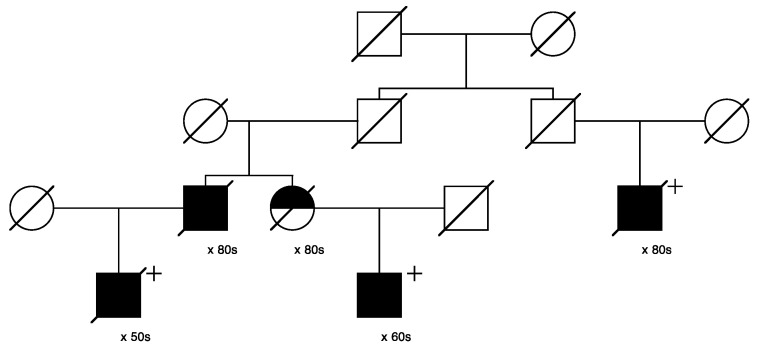
High-risk prostate cancer pedigree segregating a rare PTCH1 variant.

## Data Availability

All study subjects have relatives within the study set, which may allow identification from genetic data. Use of UPDB relationship data requires separate project application and approval. Interested researchers can contact the authors, and data will be made available to those who obtain appropriate approvals.

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
