# Peer review of "High-Risk Pedigree Study Identifies LRBA (rs62346982) as a Likely Predisposition Variant for Prostate Cancer"

_cancers, 2023, doi:10.3390/cancers15072085_

Round 1

Reviewer 1 Report

I enjoyed having the opportunity to review this study by Cannon-Albright, et al. assessing for rare predisposition variants for lethal prostate cancer.  This study leverages unique resources at the state and institution level while also utilizing other available databases for validation and also attempts to provide mechanistic rationale for their findings.  The scientific rationale and methodology are well thought out and described.  The findings are presented in an appropriate context while also appropriately highlighting the weaknesses of the study such as potential lack of external validity of findings when limiting to the Utah population.  The inclusion of supplementary data to allow for others to study is a great resource and I applaud the authors for providing this.  Studies like this are important for further exploration of rare predisposition variants in prostate cancer as they have the potential to provide diagnostic and therapeutic value.  This study is very acceptable and relevant for publication, and I have no particular concerns.

Author Response

Thank you!

No revisions made.

Reviewer 2 Report

Lisa A Cannon-Albright et al reported the LRBA (rs62346982) as a Likely Predisposition Variant for Prostate Cancer. Authors tested 51 affected cousin pairs who each died of prostate cancer. They studied the variants association in UKBiobank data. The LRBA protein struture of variant was analysied. Even if it is a interesting study, the manuscript does not show convincing data. Authors analyzed hundreds of samples but there is not a figure to show. There is not a figure to showe the whole Genome sequencing data. Furthermore, authors do not show statistical Analysis in main figures.

Author Response

Even if it is a interesting study, the manuscript does not show convincing data. Authors analyzed hundreds of samples but there is not a figure to show. There is not a figure to showe the whole Genome sequencing data. Furthermore, authors do not show statistical Analysis in main figures.

Response:

Rather than show figures summarizing WGS data, which would be common to many manuscripts, we chose instead to use figures to show the unique elements of this study, including the high-risk pedigrees, and the protein prediction modeling.  We did not identify any other figures that we deemed of interest.

Reviewer 3 Report

The authors present a discovery cohort of 51 pairs of cousins with prostate cancer diagnosed with whole genome germline sequencing to assess for both coding and non-coding risk variants. The authors found in their discovery cohort, a coding variant in LRBA gene, a gene involved in immune/CTLA4 regulation with biallelic loss of function associated with immunodeficiency syndrome. They then seek to validate the association of this variant with increase risk of prostate cancer in independent cohorts. The methodology presented is sound, though there are some areas for improvement in the results:

1) Introduction: Line 42, Recommend qualifying sentence that these genes with known associations to prostate cancer risk are NOT of equal weight or penetrance in prostate cancer risk (e.g. BRCA2 versus CHEK2).

2) Methods/Results: Recommend describing whole genome sequencing methodology in greater detail. Relying on death certificates stating "prostate cancer" as cause of death is not a reliable method of identifying "lethal prostate cancer". Without true pathologic information or clinical outcomes data, the authors should avoid stating that any risk variant found is associated with "lethal prostate cancer", but that it is associated simply with prostate cancer. Also of note in this population studied, the authors do not report any pathogenic variants in higher penetrant genes (e.g. BRCA2) which are clearly associated with increased risk of lethal prostate cancer. The excess diagnosed prostate cancers in the founder pedigree for the LRBA variant is not particularly high. Recommend a flow diagram to illustrate the starting number of variants identified (coding and non-coding) and how the variants were pared down to reach the significant finding (this is all explained in the text but would be nice to have illustrated). If possible, assess if there is any association with the variant and earlier age of disease onset. 

The protein structure data is not convincing for any mechanistic or functional link to prostate cancer risk, and does not add to this manuscript. Recommend removing it as it is largely a distraction without any preclinical validation or functional data. Furthermore, this gene has not been reported to be associated with cancer in other populations or in preclinical models, with no mention of even bi-allelic loss predisposing to increased cancer risk. If such data exists, please cite. 

The true risk associated with the LRBA variant and lethal prostate cancer warrant further evaluation in other cohorts and prospective germline testing studies with paired clinical information. In addition, this cohort is white European ancestry and more diverse populations should be evaluated for this variant (e.g. african american men who have known increased risk of lethal prostate cancer). Lastly, preclinical functional studies of the gene and altered domain associated with this variant should be performed for subsequent studies.  

Author Response

The methodology presented is sound, though there are some areas for improvement in the results:

  • Introduction: Line 42, Recommend qualifying sentence that these genes with known associations to prostate cancer risk are NOT of equal weight or penetrance in prostate cancer risk (e.g. BRCA2 versus CHEK2).

A clause has been added to the last sentence of the first paragraph of the INTRODUCTION to make this point.

  • Methods/Results: Recommend describing whole genome sequencing methodology in greater detail.

We have added details to the WGS section of METHODS.

Relying on death certificates stating "prostate cancer" as cause of death is not a reliable method of identifying "lethal prostate cancer". Without true pathologic information or clinical outcomes data, the authors should avoid stating that any risk variant found is associated with "lethal prostate cancer", but that it is associated simply with prostate cancer.

We appreciate the comment from the reviewer about their disagreement with our terminology of “lethal prostate cancer.” However, we respectfully disagree. In an epidemiological study, the only way to establish if the patient died of prostate cancer is to look at the death certificates that the treating physicians sign. An autopsy to prove the cause of death is rare, even in clinical trials or regular clinical practice. So, we would like to keep this terminology of “lethal prostate cancer” based on the death certificates. Because Utah death certificates only identify those causes of death clearly identified to contribute to death, and because all Utah prostate cancer cases analyzed here have pathologic confirmation in the Utah Cancer Registry, we believe our identification of “lethal prostate cancer” cases is accurate, although we recognize that many such cases may remain unidentified. (We have added a clause in METHODS indicating that all Utah Cancer Registry cases have pathologic confirmation).

We do agree with the reviewer about the drawing of conclusions, and although our 104 proband prostate cancer cases were thusly validated as LPrCa, our additional analyses which identified high-risk pedigrees and which assayed variants in additional, confirmed, related prostate cancer cases used only “prostate cancer” for the phenotype. For this reason, we have been conservative in our use of the “lethal” adjective when making conclusions (including in the title, abstract and throughout the manuscript).  We feel that the description provided for the LPrCa probands, and the description of the cases and further analyses clarify this point, and that we have been appropriately conservative, as is exemplified in our discussion of this specific point in the DISCUSSION (line 420-423).

Also of note in this population studied, the authors do not report any pathogenic variants in higher penetrant genes (e.g. BRCA2) which are clearly associated with increased risk of lethal prostate cancer.

The reviewer is correct that no such variants were identified in this study of 51 high risk cousin pairs.  This is not unexpected given the rare frequencies of such variants.

The excess diagnosed prostate cancers in the founder pedigree for the LRBA variant is not particularly high.

We disagree with the reviewer on this point:The founder of this pedigree .. has almost 32,000 descendants in UPDB, with a total of 230 prostate cancer cases observed and 174.9 expected (p=4.0 e-5) …”

Recommend a flow diagram to illustrate the starting number of variants identified (coding and non-coding) and how the variants were pared down to reach the significant finding (this is all explained in the text but would be nice to have illustrated).

We considered the possibility of a figure, but believe our text description is more straightforward and understandable.

If possible, assess if there is any association with the variant and earlier age of disease onset. 

We identified only 44 prostate cancer case variant carriers with age at diagnosis data. The range of age at diagnosis in these carriers was 52 to 89 years and the average and the median age at diagnosis was 70 years. The commonly communicated average age at diagnosis for prostate cancer is 69 years.  We think further data is required to clarify this question.

The protein structure data is not convincing for any mechanistic or functional link to prostate cancer risk, and does not add to this manuscript. Recommend removing it as it is largely a distraction without any preclinical validation or functional data.

We respectfully disagree with the reviewer’s recommendation. While the reviewer may find this discussion a distraction, we believe that it provides some insight on possible underlying pathogenetic mechanisms. We agree that the results presented here do not lead to a definite mechanism of pathogenesis, but at the very least highlight structural changes in the protein that may be plausible pathogenic factors. It also important to remark that the contrast of the PolyPhen results and the detailed structural analysis provided here highlights the importance of considering minor structural features that may impact protein function more that it would be expected from a global protein structure analysis or a similarity sequence approach like the one used in PolyPhen. This was an important analysis to conduct and displays important data to those interested. We do not feel it would be appropriate to exclude this data/analysis.

Furthermore, this gene has not been reported to be associated with cancer in other populations or in preclinical models, with no mention of even bi-allelic loss predisposing to increased cancer risk. If such data exists, please cite. 

We did not observe such data in addition to the data validating association with risk. But it is not unexpected that this unique study design would identify novel variants, all of which are quite rare.

The true risk associated with the LRBA variant and lethal prostate cancer warrant further evaluation in other cohorts and prospective germline testing studies with paired clinical information. In addition, this cohort is white European ancestry and more diverse populations should be evaluated for this variant (e.g. african american men who have known increased risk of lethal prostate cancer). Lastly, preclinical functional studies of the gene and altered domain associated with this variant should be performed for subsequent studies.  

We totally agree with the Reviewer and these points are included in the conclusions of our DISCUSSION section.

Round 2

Reviewer 2 Report

The authors revised the manuscript. It is an important study of predisposition variant for prostate cancer pedigree study. I would like to say that it is not enough data according to this manuscript to show the LRBA (rs62346982) as lethal prostate cancer variant, because of the number of prostate cancer patients, comparing analysis of other GWAS studies data, function of LRBA in prostate cancer etc.